# Experimental and Numerical Investigation on the Influence Factors of Damage Interference of Patch-Repaired CFRP Laminates under Double Impacts

**DOI:** 10.3390/polym15061403

**Published:** 2023-03-11

**Authors:** Zhenhui Sun, Cheng Li, Ying Tie

**Affiliations:** School of Mechanical and Power Engineering, Zhengzhou University, Science Road 100, Zhengzhou 450001, China

**Keywords:** CFRP laminates, patch repair, double impacts, damage interference, impact position

## Abstract

The impact responses of a patch-repaired carbon-fiber-reinforced polymer (CFRP) specimen under double impacts were compared to study the damage interference mechanism through the combination of experiment and numerical analysis. A three-dimensional finite element model (FEM) with iterative loading based on continuous damage mechanics (CDM) and a cohesive zone model (CZM) was employed to simulate the double-impacts testing with an improved movable fixture at an impact distance of 0 mm–50 mm. The influence of impact distance and impact energy on the damage interference was explored by mechanical curves and delamination damage diagrams of the repaired laminates. When impactors fell within the range of the patch with an impact distance of 0 mm–25 mm at a low level of impact energy, delamination damage of the parent plate caused by the two impacts overlapped, resulting in damage interference. With the continuing increase in impact distance, the damage interference gradually disappeared. When impactors fell on the edge of the patch, the damage area caused by the first impact on the left half of the adhesive film gradually enlarged, and as the impact energy increased from 5 J to 12.5 J, the damage interference caused by the first impact on the second impact was gradually enhanced.

## 1. Introduction

Carbon-fiber-reinforced polymer (CFRP) composites are widely used in the field of aerospace because of their excellent mechanical properties, such as high specific strength and high specific modulus [1]. Nevertheless, composites are vulnerable to unavoidable external load collisions and impacts during service, assembly and maintenance [2,3]. Under external load, the stability of the composite itself changes [4,5,6].A low-speed impact similar to a tool drop could cause invisible damage to the interior of the composite, such as matrix cracking, fiber fracture and delamination [7]. Those invisible damage seriously weaken the mechanical properties of the composites, making it difficult to meet the requirements of aircraft safe operation [8]. After damage occurs inside the composite structure, it is often necessary to replace or repair the damaged part. In fact, due to the complexity and large size of the composite structure, it is often not feasible to completely replace the damaged parts [9]. In the past ten years, adhesive repair technology has become a commonly used repair technology to restore the mechanical properties of composite materials due to its simple operation process, lower equipment requirements and relatively short repair time [10,11,12]. The commonly used adhesive repair technology can be divided into patch repair and scarf repair. Although scarf repair can well restore the mechanical properties of damaged structures, a large number of the materials of undamaged parts need to be excavated in the repair process [13]. Compared with scarf repair, a patch repair operation can also restore the mechanical properties of damaged specimens. Moreover, it is simpler and feasible [14]. Therefore, this paper mainly studies the impact behaviors of patch-repaired composite structures.

The repaired composite structure often suffers impacts of low-speed and low-energy objects during service; most of these low-speed impacts have the characteristics of occurring at multiple times and having multi-point effects, resulting in damage interference and accumulation of the structures. Therefore, the impact resistance of the repaired structure has become the focus of many scholars. So far, the single-impact performance of the repaired structure has been extensively studied. Liu et al. [15] conducted a series of impact experiments to obtain the critical impact energy of adhesive layer damage. They further studied the internal failure mechanism of the repaired structure under impact load. Ivañez [16] studied the effect of the impact point on the damage tolerance of repaired laminates through experiments. They found that the CAI strength after impact at the center of the repaired specimen was significantly higher than the residual strength after impact at the edge of the patch. Jiang [12] studied the relationship between fiber orientation and impact resistance of repaired laminates under impact load. For A-type composite materials, the fiber orientation of the patch affected the stress direction of each layer of the composite material. In summary, the impact resistance of the patch-repaired structure is affected by the impact energy, impact position and patch fiber direction. Additionally, the size and shape of the patch are also key factors affecting the impact resistance of the repaired structure [17]. In practice, the repaired structure is subjected to more than one impact during service [18,19]. The structural damage caused by each impact will gradually accumulate with the increase in the number of impacts, causing more serious failure of the structure [20]. Coelho [21] studied the impact response of single-sided patch-repaired and double-sided patch-repaired structures under multiple impacts through experiments. They found that the double-sided-patch-repaired structure had a longer impact fatigue life because its overall stiffness was larger than the single-sided patch-repaired structure.

As a special case of repeated impacts, the impact resistance of composite structures under double impacts has attracted the attention of many scholars. Liao [22,23] studied the effect of impact position on the impact response of composite laminates under double impacts through experiments. They recorded the force–time curves and force–center displacement curves during each test. Experimental results showed that the degree of interference between two impacts can be characterized by the maximum center displacement. Zhou [24] used the same energy to impact carbon-fiber-reinforced polymer specimens at two locations at the same distance from the center. They recorded the mechanical response and various types of damage caused by the two impacts. The experimental results showed that the maximum displacement of the specimen during the first impact reflected the degree of difficulty of the specimen deformation. The difficulty of the deformation had an important influence on the damage shape and damage area of the specimen.

Compared with common composite laminate specimens, patch-repaired structures involve the nonlinearity brought by the patch and adhesive layer. It is difficult to obtain mechanical data such as stress distribution and damage distribution within the structure only through experimental research on the repeated impact performance of the repaired structure. Moreover, the cost of the experiments is relatively high, and the experiment results are also subject to greater chance. There is little research literature about patch-repaired structures with repeated impacts. However, the damage evolution and damage interference mechanism of patch-repaired structures under double impacts need more detailed research. It is necessary to establish a finite element model (FEM) to study the mechanical response and internal damage evolution of the repaired structure under repeated impacts.

In this paper, the damage interference mechanism of a patch-repaired structure under the double impacts through experiment and simulation is investigated to derive the damage mechanism of the repair structure by analyzing the influence relationship between the first impact and the second impact.

## 2. Experimental Procedure

### 2.1. Specimen Preparation

As shown in Figure 1a, the test specimen used in this work consists of three parts: adherend (parent plate), patch and adhesive film. The parent plate and the patch with carbon fiber/epoxy composite laminate, model T300/7901, were provided by Shandong Weihai Guangwei Corporation in China. T300/7901 means that the composite laminate consists of carbon fiber of the T300 type and resin of the 7901 type. The stacking sequence of the parent plate with dimensions of 150 mm × 100 mm × 3.6 mm was [(0_3_/90_3_)_2_]_S_. The parent plate contained 24 single layers and the thickness of each single layer was about 0.15 mm. A vertical circular hole with a radius of *r* = 12.5 mm (impactor radius) was cut with a water jet in the middle of the parent plate to simulate the damage of the parent plate caused by accidents. In order to avoid reading ambiguity, the x-direction was defined as the length direction of the specimen, which was also the 0° fiber direction of the specimen. The y-direction was defined as the width direction of the specimen and the 90° fiber direction. The specified z direction was the thickness direction of the specimen.

The damaged parent plate and the patch were connected by the adhesive film to achieve the repair of the damaged parent plate. According to references [17,25,26], the size, thickness and shape of the patch affects the repair effect. A better repair effect can be obtained when a patch with a radius of 2 *r*–2.5 *r*, a circular shape and half the thickness of the parent plate is used to repair the damaged parent plate. Therefore, circular patches with a radius *R* twice as large as the damaged hole, half the thickness of the parent plate and a stacking sequence of [(0/90)_3_]_S_ were used to repair the damaged laminates. Figure 1a shows the patch-repaired structure obtained. The part marked yellow in the structure is the LJM-200 type thermosetting epoxy resin adhesive film (provided by Shandong Weihai Guangwei Corporation in China) cured at a medium temperature (120 °C), with a surface density of 200 g/m^2^ and a thickness of about 0.1 mm. The patch and the parent plate were connected with the adhesive film after aligning the patch center with the parent plate center; the obtained specimens were then heated in a SG-XL1200 (provided by Shanghai Institute of Optics and Precision Machinery in China) type box furnace at a temperature of 120 ° C for 2 h to achieve curing. All specimens were non-destructively tested by AX8200 X-ray fluoroscopy inspection equipment (provided by Wuxi Rilian Technology Co., Ltd in China) before repairing to ensure that there was no damage in the specimens except for prefabricated damage holes. The front and back of the obtained specimen are shown in Figure 1c and Figure 1d, respectively.

### 2.2. Low-Velocity Double-Impacts Test Platform

As shown in Figure 2, a low-velocity double-impacts platform for patch-repaired composite laminate, including an XBL-300 drop weight impact test machine produced by Changchun Kexin Co., Ltd in China and the supporting test data acquisition device, was built according to ASTM D7136/D7136M-15 (provided by the American Society for Testing and Materials in West Conshohocken, PA, USA) [27]. From the top to the bottom, the impactor, anti-secondary impact device, improved specimen fixture and high-speed camera constituted the main part of the XBL-300 drop weight impact tester. The dynamic force sensor (PCB 208C05), produced by PCB Piezotronics Inc. in New York, NY, USA, and the supporting data acquisition system constituted the data acquisition device for the experiments. The dynamic force sensor was a piezo-electric load cell, which could transform the piezo-electric signal into a force–time signal. As shown in Figure 2b,c, the improved fixture was composed of two movable baffles, four fixed bolts, four locating studs and the upper and the lower fixtures. The two movable baffles were inserted into the two square grooves, respectively opened by the upper fixture and the lower fixture, which limited the front and rear positions of the specimen. The fixed bolts connected the upper and the lower fixtures together, which fixed the upper and lower positions of the specimen. The locating studs passed through the guide groove of the lower fixture, and were connected to the impact testing machine, so that the lower fixture could move back and forth through the guide groove. The distance of the front and back movement could be determined by observing the reference dimension line set on the side of the lower fixture. The forward and backward movement of the lower fixture could drive the specimen to move forward and backward. Since the impact point of the impactor was fixed in the horizontal direction, the impact on the specimen at different positions in the forward and backward direction could be realized by changing the forward and backward positions of the lower fixture. The high-speed camera (i-speed221) recorded the initial impact velocity. More importantly, the guide rail used to guide the impactor to make a free-fall motion was lubricated regularly to minimize friction loss.

### 2.3. Low-Velocity Double-Impacts Test Process

As shown in Figure 1b, each specimen was subjected to double impacts during the impact process, and the two impacts were not carried out at the same time. Two impact points were symmetrically located on the left and right of the midpoint; the distance between impact point and midpoint was d. In order to distinguish the experiments, a notation *E-d-1/2* was used to represent a specific pair of experiments, where *E* represented the impact energy and *d* represented the impact distance of the first (*1*) and second (*2*) impact, respectively. It should be noted that the two impact points were both at the midpoint of the specimen when *d* was taken as 0 mm. Impact test parameters of patch-repaired double-impacts specimens are listed in Table 1. 

Before the impact, the position of the improved fixture relative to the impactor (2.5 kg weight, 12.5 mm-radius hemispherical punch at the lower end) was adjusted according to the impact distance, and then the test specimen was put into the fixture and fixed. The height between the impactor and the upper surface of the test specimen was calculated according to the impact energy, and the impactor was presented at the corresponding height. The impact tester released the impactor onto the specimen after the impact started. The force curve between the impactor and the specimen was collected by the data acquisition system (the acquisition frequency was 100 kHz). Each specimen was subjected to two impacts, and the distance between the two impacts was 0 mm, 12.5 mm and 50 mm, respectively. In these three pairs of experiments with the same impact energy of 10 J, the impactor fell on the center of the patch, the edge of the patch and the edge of the laminate, respectively. The specific test process is shown in Figure 3.

### 2.4. Low-Velocity Double-Impacts Test Data Processing

Before each impact test, the impactor was dropped from the set height to check the initial impact velocity $v_{0}$ through the high-speed camera. If the velocity was not correct, the set height was corrected according to the expected initial velocity. From the moment the impactor contacted the specimen, the data acquisition system collected the impact force *F* and impact time *t* transmitted from the sensor. The absorbed energy curves and force–displacement curves of the repaired specimen could be obtained by Equations (1)–(3):(1)$$v\left(t\right)=v_{0}-\frac{1}{m}\int_{0}^{t}Fdt$$
(2)$$d\left(t\right)=\int_{0}^{t}v\left(t\right)dt$$
(3)$$E_{a}\left(t\right)=\frac{m\left(v_{0}{}^{2}-v{\left(t\right)}^{2}\right)}{2}$$
where *v*(*t*) was the impact velocity of the impactor at time point *t*, *d*(*t*) was the central displacement of the impactor at time point *t*, *m* was the weight of the impactor and$\;E_{a}\left(t\right)$ was the energy absorbed by the specimen at time point *t*. Figure 4 shows the calculation flow of the force–displacement curve and the absorbed energy curve. Furthermore, the key impact parameters could be extracted from the force–time curve, absorbed energy curve and force–displacement curve. These parameters included peak force ($F_{p}$), maximum displacement ($d_{max}$), impact time, peak energy moment (PEM) and absorbed energy ($E_{a}$). In order to analyze the interference relationship between the first impact and the second impact, the difference of$\;F_{p}\left(\delta_{f}\right)$, the difference of the PEM ($\delta_{t}$), the difference of $d_{max}\left(\delta_{d}\right)$ and the difference of impact time ($\delta_{it}$) between the first impact and the second impact were obtained through those curves.

## 3. Numerical Model

### 3.1. Intralaminar Damage and Interlamiar Damage

The force–time curves in the impact process were obtained through the impact experiments and data acquisition, and then the impact energy–time curves were derived. In addition, the surface damage of the specimens after low-velocity impacts can also be analyzed. However, it is difficult to observe the evolution of the internal damage of the specimens through impact experiments. Additionally, the randomicity of the test increases the cost of the test. Therefore, scholars have tried to use the numerical analysis method to build a finite element model (FEM) to simulate the low-velocity impact damage process of repaired specimens in recent years [28,29]. The damage of composite laminates mainly includes matrix damage, fiber damage and delamination damage in impact experiment analysis [13]. In finite element analysis, matrix damage and fiber damage are considered as intralaminar damage, which are simulated based on continuous damage mechanics (CDM) [28,30]. The delamination damage is considered as interlaminar damage, and the separation process of a single layer with different layer angles was simulated based on the cohesive zone model (CZM) [31]. Further details of these composite damage models can be found in our previous studies [20].

### 3.2. Establishment of the FEM for Double Impacts

According to the geometric parameters of the laminate and the test platform in Section 2.1, the model was built using Abaqus/Explicit [32] software. The user material subroutine VUMAT was used to simulate the material damage and evolution, and a three-dimensional FEM of the double-impacts patch-repaired structure was established. According to the actual test configuration, the double-impacts FEM in Figure 5b consisted of the damaged parent plate, the patch, adhesive film, impactors and the fixture. The interface layers were introduced into the parent plate and the patch to simulate the possible delamination in the repaired structure, as shown in the red part of Figure 5b. After the first impact, the impactor for the first impact is removed and the impactor for the second impact is made to contact with the specimen. The impactors for each impact had an initial velocity corresponding to the impact energy when they contacted the specimen.

Before the first impact started, the lowest point of the hemispherical 1st impactor was brought into contact with the corresponding impact point. While the lowest point of 2nd impactor was kept at a certain distance from the corresponding impact point. During the first impact, the 1st impactor contacted the specimen with its speed gradually decreasing to zero and then increasing due to the rebound effect. During this period, the 2nd impactor maintained its initial velocity and approached the repaired specimen downward in the z direction. After the first impact was completed, the 1st impactor moved away from the parent plate at the remaining speed. Meanwhile, the lowest point of the 2nd impactor just touched the corresponding impact point. After the first impact, the 2nd impactor impacted the specimen at a given initial speed for a second impact. The degrees of freedom of the impactors were restricted except for the degree of freedom of movement along the z direction during the first and the second impact. The constrained state of the upper and lower fixtures is also described in Figure 5c. The specimen was clamped by the upper and lower fixtures during the impact process, and all degrees of freedom of the upper and lower fixtures were restricted.

An interface layer was introduced between each composite layer of the damaged parent plate and the patch to simulate delamination. An adhesive film was added between the damaged parent plate and the patch. In order to make the finite element calculation easy to converge and ensure the continuity of the displacement and stress of each element in the FEM, the nodes between the composite layer, interface layer and adhesive film in the model were shared [20], as shown in Figure 5d. The premise of sharing nodes depended on reliable and accurate mesh division. Due to the computing power of the computer, mesh refinement was applied to the impacted area of the model. After repeated verification of the element size of the FEM, the element sizes of 1 mm, 1.25 mm and 5 mm were employed for the parent plate, patch and adhesive film, as shown in Figure 5e. In the finite element calculation, the general contact algorithm was used to deal with the possible contact. In general contact, the normal contact characteristic was hard contact, and the tangential contact characteristic was set with a friction coefficient of 0.3.

Through experimental observation, the upper and lower fixtures and impactors had almost no deformation due to the short impact time during the impact process. Therefore, the fixtures and impactors were set as rigid bodies in the finite element analysis. In addition, the friction force of the impactors falling along the slide rail was almost zero in the experiments. Without considering the friction force, the kinetic energy lost by the impactors can be regarded as the damage dissipation energy of the specimen in the finite element calculation. As for the element types used in each part of the specimen, 3D, zero-thickness and 8-node cohesive elements (type: COH3D8) were applied to the interface layers and adhesive film to simulate the patch peeling and delamination. Additionally, 3D, 8-node solid elements were employed for fixtures, impactors and composite layers. According to the actual impact process time, the time of each impact was set as 6 ms in the finite element analysis. The time increment in the numerical calculation adopted the system default automatic increment; the linear bulk viscosity coefficient and quadratic viscosity coefficient in the damping coefficients also adopted the default values (0.06 and 1.2). The 3D FEM of the double-impacts patch-repaired specimen was divided into 63,596 elements. It took about 26 h to complete a numerical calculation using a computer with an Intel (R) Xeon (R) 24-core CPU provided by Intel Corporation in Santa Clara, CA, USA.

Delamination damage often occurs between composite layers with different ply angles, so several ply layers with the same ply angle can be regarded as a composite layer. An interface layer was introduced between composite layers with different ply angles. According to the above division method, the parent contained 7 composite layers and 6 interface layers, while the patch contained 11 composite layers and 10 interface layers. More details are presented in Figure 6.

### 3.3. Material Properties

The properties include the parameters used to calculate intralaminar damage, the interlaminar damage of the CFRP laminate and the parameters of the adhesive film. The mechanical properties of each part of the T300/7901CFRP laminate repaired by the patch can be obtained from Ref. [20], as shown in Table 2.

## 4. Results

### 4.1. Experimental Results and Discussion

In this section, the impact energy of 10 J was used for three pairs of specimens at impact distances of 0 mm, 12.5 mm and 50 mm. Each pair of double-impacts experiments was repeated five times to eliminate the interference caused by random errors. Under the stable experiment environment and skilled manual operation, the force curve obtained by each group of patch-repaired specimens fluctuated between 3% and 5%.In order to distinguish the experiments, a notation *E-d-1/2* was used to represent a specific pair of experiment, where *E* represents the impact energy and *d* represents the distance from the impact point of the first impact (*1*) and the impact point of the second impact (*2*) to the center of the test specimen, respectively. It should be noted that the two impact points were both at the midpoint of the specimen when *d* was taken as 0 mm.

#### 4.1.1. Behavior of Patch-Repaired Composites under Double Impacts

According to Equations (1)–(3) in Section 2.4, the force–time curves obtained by the force sensor can be converted into the impact absorbed energy–time curves. For the convenience of description, the force–time curve and the impact absorbed energy–time curve are respectively referred to as the force curve and the impact energy curve. Figure 7 shows the force curve and the corresponding impact energy curve of the double-impacts specimen under three impact distances. As shown in Figure 7a, when the impact distance was 0 mm, the force curve caused by the first impact showed a fluctuating and rising trend due to the damage of the specimen in the rising stage of the force value compared with the second impact. Compared with the first impact, the force curve was smooth at the rising stage due to the steady release of energy dissipation. The force curve then reached the peak force. Due to the compaction effect inside the composites [33], the peak force of the second impact was 4658.9 N higher than the 4500.4 N of the first impact. As for the impact energy curves of the two impacts, it was observed that the energy absorbed by the specimen in the second impact was 2.61 J, which was less than the 3.46 J in the first impact. In the second impact, the internal damage of the composite expanded further, but, due to the compaction effect, the repaired specimen showed better impact resistance in the second impact.

Figure 7b shows the force curves and the resulting impact energy curves of the specimens under double impacts when the impact distance was 12.5 mm. It can be seen that, during the first impact, the force curve fluctuated greatly in the rising stage, and the specimen suffered serious internal damage during this process. During the second impact, the force curve was relatively smooth in the rising stage, but the force curve still had a lot of jitter in the whole process. Combined with the impact energy curve obtained from the two impacts, it can be seen that the repaired structure absorbed more energy in both impacts. The energy absorbed by the second impact was less than that of the first impact, and the time to complete the second impact was also higher than that of the first impact, indicating that the structural stiffness of the repaired structure decreased after the first impact, which increased the impact time. When the impact distance was 12.5 mm, the impact points of the first impact and the second impact were close, and the damage caused by the first impact overlapped with the damage caused by the second impact, reducing the energy absorbed by the repair structure during the second impact.

The force curves and impact energy curves caused by two impacts on the repaired specimen when the impact distance was 50 mm are shown in Figure 7c. Unlike the specimens with impact distances of 0 mm and 12.5 mm, the force curve showed a sudden downward trend at about 0.0005 s. At this time, it can also be found through visual observation that serious damage occurred in the repaired structure, and the energy of the impactor was almost entirely absorbed by the repaired specimen according to the impact energy curve. The peak force value, impact time and force curve fluctuation of the force–time curves of 10-50-1 and 10-50-2 were almost consistent. The absorbed energy curves of 10-50-1 and 10-50-2 were basically consistent. When the impact distance was 50 mm, the two impact points were 100 mm apart. Such a large distance made the correlation between the first impact and the second impact almost zero.

#### 4.1.2. Damage Profiles

The surface damage and internal damage of the patch-repaired specimen after double impacts had different performances with the change in impact conditions. In this section, we attempted to observe the surface morphology and internal damage of the specimen through the ultra-depth-of-field digital microscope to analyze the damage rule of the specimen at different impact distances. It was difficult to observe the internal damage of the specimen directly. In this section, the impacted specimen was cut along the x-direction with a water jet, and then the damage distribution of the cut section of the specimen was observed with the microscope, as shown in Figure 8d, Figure 9d and Figure 10d.

Through the observation and analysis of Figure 8a,b, it can be seen that when the impact distance was 0 mm, almost all damage occurred on the patch. There was a small range of fiber fracture and matrix cracks centered on the impact point on the front of the patch, and the matrix cracking damage was mainly on the back of the patch. Section A-A reflects the damage inside the patch near the impact point. It can be seen that the patch had obvious delamination damage under the impact load. Section B-B reflects the damage of the adhesive film. There was a small range of adhesive film cracking at the right position of the patch.

As shown in Figure 9, the two impact points were so close that almost all damage also occurred on the patch at the impact distance of 12.5 mm. There was a small range of fiber fracture and matrix cracking centered on the two impact points on the front of the patch, and the matrix cracking damage was mainly on the back of the patch. The D-D section reflects the damage inside the patch near the impact point of the first impact. Under the double impact load, there was obvious delamination damage in the middle of the patch. The C-C section reflects the damage of the adhesive film. At this time, the adhesive film was completely torn and the patch was separated from the parent plate. Seen from Figure 10, the two impact points were so far apart that the patch was little damaged at the impact distance of 50 mm. Matrix cracking occurred only near the impact point on the front of the parent plate. Section F-F reflects the damage of parent plate near the impact point. It can be seen that there was serious delamination damage inside the parent plate.

### 4.2. Simulation Results and Discussion

Through a large number of tests and high-precision detection instruments, the damage distribution and various impact data of the impact specimen can be accurately grasped, but this often consumes a lot of human and material resources. With the application of FEM, the cost of the scientific research was reduced, and the damage inside the specimen was easily obtained through numerical simulation. In order to apply the FEM to simulate the impact process, the correctness of the model needed to be checked first. On this basis, Section 4.2 discusses the influence of impact distance and impact energy on the damage interference between the two impacts of the double-impacts patch-repaired specimens.

#### 4.2.1. Experimental Validation of the Numerical Model

When the impact distance was 0 mm, the impact points of two impacts fell in the middle of the patch. The comparison of the experiment and simulation results of the front and back damage of the repaired specimen after double impacts is shown in Figure 11c. Since the patch bore the main impact load, the damage mainly appeared on the patch. The damage morphology of the experiment and simulation was consistent. On the front of the patch, the fiber damage was *X*-shaped. On the back of the patch, there was no fiber damage, only matrix damage extending along the long axis. Figure 11a,b shows the force curves and impact energy curves obtained from simulations and experiments during the first and second impact. The force curves and impact energy curves had a good fitting relationship.

Because the impact points of the two impacts were staggered, there were two concentration points of fiber damage and matrix damage on the front of the specimen when the impact distance was 12.5 mm. Compared with the impact distance of 0 mm, the matrix damage still extended along the long axis at the back of the patch, but the damage area expanded. More comparison results of experiment and simulation are shown in Figure 12.

In order to better compare the impact response results of the experiment and the numerical analysis of the specimen under two impact distances, the impact parameters, such as peak force and absorbed energy value, were extracted from the force curves and impact energy curves in Figure 11 and Figure 12. The comparison results of the impact parameters of the experiment and numerical analysis are summarized in Table 3. It can be seen that the comparison errors of each data were within 7.89%. Observation of Figure 11c and Figure 12c shows that the damage morphology of the experiment and numerical analysis were also quite consistent. The correctness of the FEM was proved by the comparison of impact parameters and damage morphology.

#### 4.2.2. Effect of Impact Distance on the Damage Interference of Patch-Repaired Specimens under Double Impacts

The correctness of the numerical calculation was demonstrated previously. This Section 4.2.2, studies the damage behavior of the patch-repaired specimens under double-impacts when the impact distance increased from 0 mm to 50 mm. The selected impact distances were 0 mm, 12.5 mm, 25 mm, 37.5 mm, and 50 mm, respectively.

##### Impact Force–Time Curves

Figure 13a–o shows the force–time curves, impact energy curves and force–displacements curves at the impact distances of 0 mm–50 mm, with the impact energy of 10 J. By observing the force–time curves under five impact distances, it can be found that $\delta_{f}$ reached the maximum at the impact distance of 25 mm, which was the edge of the patch. The damage caused by the first impact affects the impact behavior of the second impact. When the impact distance was 0 mm, the slope of the rising section of the force–time curve of 10-0-2 was greater than that of 10-0-1. The impact points of 10-0-2 and 10-0-1 fell at the same point, which made the damage propagation of 10-0-2’s patch-repaired specimen stable and the force–time curve smooth. There was little difference between the peak forces of the first impact and the second impact when the impact distance was 0 mm or 12.5 mm, and when the impact distance was 37.5 mm or 50 mm. However, the peak force of the second impact was significantly less than that of the first impact at an impact distance of 25 mm. The first impact weakened the stiffness of the repaired specimens and reduced the peak force of the second impact at an impact distance of 25 mm. In general, the difference between the peak forces of the first impact and the second impact did not show certain regularity with the increase in impact distance, so it was difficult to describe the interference between the first impact and the second impact by the peak forces.

##### Absorbed Energy Curves

The absorbed energy curves of the repaired specimen under five impact distances are shown in Figure 13b,e,h,k,n. In the cases that the impact distance was 0 mm or 12.5 mm, and the impact points fell inside the patch, it can be concluded that the energy absorbed by the specimen during the first impact was greater than that of the second impact, indicating that the damage caused by the first impact accumulated during the second impact and affected the second impact. Combined with the delamination damage diagrams of the repaired specimen in Figure 14a, the delamination damage caused by the first impact and that caused by the second impact overlapped, so that the energy absorbed by the repaired specimen during the second impact was less than that of the first impact. When the impact distance was 25 mm, the impact point fell on the edge of the patch. The energy absorbed by the repaired specimen during the second impact was close to that absorbed during the first impact. According to Figure 14a, the adhesive film damage caused by the first impact and that cause by the second impact at the impact distance of 25 mm was different. Additionally, the trend of the force–time curve of 10-25-1 was also significantly different from that of 10-25-2. These two differences show that the first impact caused damage interference with the second impact. Therefore, the interference relationship between the first impact and the second impact cannot be evaluated simply by the difference of absorbed energy between the first impact and the second impact. When the impact distance was 37.5 mm or 50 mm, the impact point fell on the parent plate. Due to the large impact distance, the interference between the first impact and the second impact was weak, and the energies absorbed by the repair specimen during the first impact and the second impact were almost the same.

In addition, *δ_t_* also changed with the increase in the impact distance. When the impact distance increased from 0 mm to 12.5 mm, the difference of PEM decreased from 0.13 ms to 0.08 ms, indicating that the interference caused by the first impact on the second impact was gradually weakened. When the impact distance increased from 12.5 mm to 25 mm, the peak energy difference of PEM increased from 0.08 ms to 0.20 ms, indicating that the interference between the first impact and the second impact was enhanced. Combined with the delamination damage diagrams of the repaired specimen in Figure 14a, it can be seen that the adhesive film damage of 10-25-1 and that of 10-25-2 was significantly different. There was a large section of continuous adhesive film damage at the left end of 10-25-1’s adhesive film, so that the left end of the patch of 10-25-2 could not bear the impact load during the impact process. This weakened the overall impact resistance of the repaired specimen, resulting in the increase in the difference of PEM. Moreover, with the increase in the impact distance from 37.5 mm to 50 mm, the difference of PEM was basically the same, indicating that the interference between the first impact and the second impact was weak.

##### Impact Force–Displacement Curves

The force–displacement curves of the repaired specimen under five impact distances are shown in Figure 13c,f,i,l,o. It can be found that the maximum displacement among the five impact distances, that of 10-0-2, reached 5.162 mm. It can be explained that, for specimen 10-0, the impactor completely fell on the center of the patch. Because the stiffness of the repaired specimen at the center of the patch was weak, the maximum displacement was reached there. Then, when the impact distance increased from 0 mm to 25 mm, the maximum displacement caused by the second impact was always greater than that of the first impact, which indicated that the damage caused by the first impact weakened the overall stiffness of the repaired specimen, making the specimen more prone to deformation during the second impact.

When the impact distance continued to increase to 50 mm, due to the weak interaction between the first impact and the second impact, the force–displacement curves of the two impacts maintained a similar trend. Furthermore, $\delta_{d}$ was extracted to analyze the displacement interference degree between the first impact and the second impact. It was not difficult to find that the maximum displacement difference decreases with the increase in impact distance. The displacement interference between the first impact and the second impact decreased gradually with the increase in impact distance.

##### Delamination Damage

The appearance of delamination damage inside the specimen under impact load was the main reason for the decline of the mechanical properties of the specimen. Figure 14a presents the first-impact and second-impact delamination damage diagrams for the patch, adhesive film and the parent plate under five impact distances. It can be seen from the figure that, when the impact distance was 0 mm, the patch bore the main impact load and the delamination damage mainly occurred on the patch, while the parent plate did not exhibit delamination damage. When the impact distance was 12.5 mm, both the first impact and the second impact caused delamination damage to the parent plate and the patch. When the impact distance was 25 mm, the first impact seriously damaged the left half of the adhesive film, causing the patch to partially fall off. When the impact distance was 37.5 mm or 50 mm, the parent plate bore the main impact load, and delamination mainly occurred on the parent plate. The impact parameters were extracted from the individual curves in Figure 13. These impact parameters included peak force, maximum displacement, impact time, PEM, absorbed energy and delamination damage projected area (DDPA), as listed in Table 4. The impact parameter curves were made according to Table 4, as shown in Figure 14b. As can be seen from Figure 14b, the PEM, impact time and maximum displacement curve have the same gradual decreasing trend. The difference curves between the first impact and the second impact of these three parameters are shown in Figure 14c. The $\delta_{t}\;$curve and $\delta_{it}$ curve reflect the first increasing and then decreasing trend of damage interference with impact distance. It can also be seen from Figure 14b that, as the impact distance increased from 0 mm to 37.5 mm, the DDPA of the parent plate showed an overall increasing trend. Therefore, the farther away from the center of the patch, the more prone the repaired specimen was to delamination damage due to insufficient protection of the patch under impact load.

#### 4.2.3. Effect of Impact Energy on the Damage Interference of Patch-Repaired Specimens under Double Impacts

When the impactor impacted the edge of the patch, the damage of the repaired specimen under double impacts was complicated. Therefore, in this section, the impact distance was selected as 25 mm, the impact energy was increased from 5 J to 15 J and the interference relationship between the first impact and the second impact under five impact energies was studied.

The force–time curves under five impact energies are shown in Figure 15a,d,g,j,m. It can be found that $\delta_{f}$ reached the maximum value when the impact energy was 10 J. When the impact energy increased from 5 J to 10 J, $\delta_{f}$ basically showed a gradually increasing trend, and the peak force of the first impact was larger than that of the second impact. The damage to the repaired specimen caused by the first impact weakened the impact resistance of the specimen, so that the peak force of the second impact was smaller than that of the first impact. However, when the impact energy was 15 J, the peak force of the second impact was larger than that of the first impact. Figure 16 shows the contact between the impactor and the patch at the first impact and the second impact when the impact energy was 15 J. As shown in Figure 16, the first impact separated the left half of the patch from the parent plate. Due to the lack of constraint on the left, the patch tilted to the right, which increased the contact area between the impactor and the patch in the second impact. This resulted in a larger peak force from the second impact than from the first impact. Therefore, the interference relationship between the first impact and the second impact cannot be evaluated using only the peak force.

The absorbed energy curves of the repaired specimen under five impact energies are shown in Figure 15b,e,h,k,n. The energy absorbed by the second impact was higher than the first impact when the impact energy was small (5 J–7.5 J). Combined with Figure 17a, the increased DDPA (177 mm^2^ and 312 mm^2^) of the adhesive film caused by the second impact was greater than the DDPA (168 mm^2^ and 289 mm^2^) of the adhesive film caused by the first impact. This caused the specimen to absorb more energy during the second impact. Combined with Figure 17a, the increased DDPA (252 mm^2^ and 402 mm^2^) of the adhesive film caused by the second impact was less than that caused by the first impact (589 mm^2^ and 681 mm^2^) when the impact energy increased from 10 J to 12.5 J. According to the above analysis on the increase in impact energy from 5 J to 7.5 J, the energy absorbed by the first impact should have been greater than that absorbed by the second impact. In fact, there was not much difference between the energy absorbed by the specimen at the first impact and that at the second impact. Through further observation, it was found that, when the impact energy was 10 J or 12.5 J, the left half of the patch was basically separated from the parent plate. Because the left half of the patch was out of restraint, the increase in adhesive film damage during the second impact was needed to absorb more energy. As a result, the specimen absorbed almost the same energy in the first impact and the second impact when the impact energy was 10 J or 12.5 J.

In addition, $\delta_{t}\;$also changed with the increase in the impact energy. When the impact energy increased from 5 J to 12.5 J, $\delta_{t}$ increased gradually. When the impact energy was 15 J, $\delta_{t}$ was almost zero. Combined with Figure 17a, when the impact energy increased from 5 J to 15 J, the DDPAs of the adhesive films caused by the first impact were 168 mm^2^, 289 mm^2^, 589 mm^2^, 681 mm^2^ and 710 mm^2^, respectively (Table 5). The increased DDPAs of the adhesive films after the second impact were 177 mm^2^, 312 mm^2^, 252 mm^2^, 402 mm^2^ and 730 mm^2^, respectively. When the impact energy was 15 J, the DDPA of the adhesive film caused by the first impact and the second impact was basically the same, which led to $\delta_{t}$ being almost zero.

Similar to the changing trend of $\delta_{t}$ with the impact energy, $\delta_{d}$ increased with the increase in the impact energy from 5 J to 12.5 J; however $\delta_{d}$ was close to zero when the impact energy increased to 15 J. This showed that, due to the large impact energy, large damage occurred in the first impact and the second impact, so that there was almost no interference between the first impact and the second impact. As can be seen from Figure 17b, the PEM, impact time and maximum displacement curves all show a trend of increasing first and then tending to be stable. The difference curves between the first impact and the second impact of these three parameters are shown in Figure 17c. The$\;\delta_{t}\;$ and $\delta_{d}$ curves reflect the first increasing and then approaching-zero trend of damage interference with impact energy. Based on Figure 14c and Figure 17c, at different impact distances and impact energies, compared with $\delta_{d}$ and $\delta_{it}$,the change trend of $\delta_{t}$ was more consistent with the impact damage interference, so $\delta_{t}$ can be used to characterize the degree of interference between the first impact and the second impact. It can also be seen from Figure 17b that, as the impact energy increased from 0 J to 15 J, the overall DDPA of the parent plate tended to increase. Therefore, the effect of increasing impact energy on the damage of repaired specimens was significantly expressed in terms of delamination area.

## 5. Conclusions

A low-velocity double-impacts experiment was carried out on a patch-repaired CFRP specimen using a low-velocity impact test platform equipped with an improved movable fixture. Based on the CDM and CZM, a three-dimensional FEM was built to carry out numerical analysis on the basis of the test, and the double-impacts damage behavior of the repaired specimen was successfully simulated. Finally, the damage interference mechanism under different impact distances and different impact energy levels was further studied by numerical analysis. This paper puts forward the following conclusions:
An FEM of double impacts was established based on the CDM and CZM. The comparison errors of the impact data were within 7.89%. The damage morphology of the experiment and numerical analysis were also quite consistent. The correctness of the FEM was proved by the comparison of impact parameters and damage morphology.When the distance between the impact points of the first impact and the second impact increased from 0 mm to 25 mm (impactors fell on the patch) at a low level of impact energy (10 J), the damage caused by the first impact overlapped with the damage caused by the second impact, reducing the energy absorbed by the repair structure during the second impact. When the distance between the first impact and the second impact increased to a certain extent (e.g., at 50 mm, the impactors fell outside the patch), and there was little correlation between the two impacts.When impactors fell on the edge of the patch (at an impact distance of 25 mm), the damage interference caused by two impacts was complex. The $\delta_{t}$ curve and $\delta_{d}$ curve basically showed a gradually increasing trend when the impact energy increased from 5 J to 12.5 J at an impact distance of 25 mm. The interference between the first impact and the second impact gradually increased with the rise of impact energy.As the impact distance increased from 0 mm to 37.5 mm at an impact energy of 10 J, the DDPA (0–1622 mm^2^) of the parent plate showed an overall increasing trend. Therefore, the farther away from the center of the patch, the more prone the repaired specimen was to delamination damage due to insufficient protection of the patch under the impact load. As the impact energy increased from 0 J to 15 J at an impact distance of 25 mm, the overall DDPA of the parent plate tended to increase.The PEM, impact time and maximum displacement curves had the same changing trend. At different impact distances and impact energies, compared with $\delta_{d}$ and $\delta_{it}$, the change trend of $\delta_{t}$ was more consistent with the impact damage interference, so $\delta_{t}$ was be used to characterize the degree of interference between the first impact and the second impact.

## Figures and Tables

**Figure 1 polymers-15-01403-f001:**
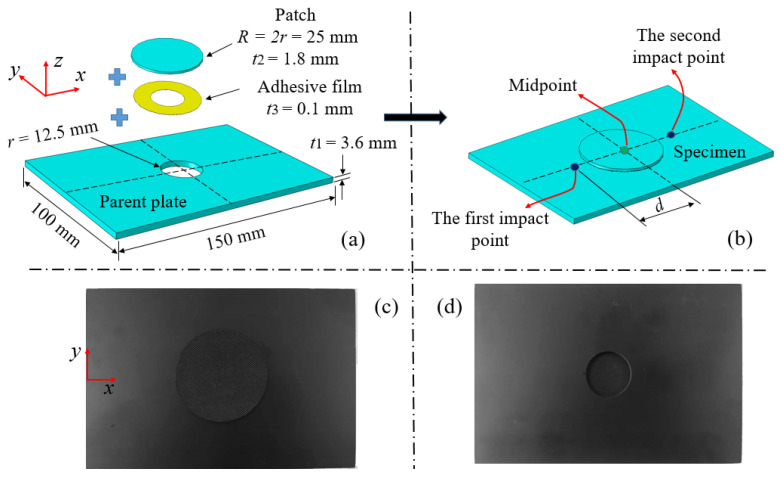
Geometry information of (**a**) the parent plate and (**b**) the geometry configuration of the patch-repaired specimen and a photograph of (**c**,**d**) the patch-repaired specimen.

**Figure 2 polymers-15-01403-f002:**
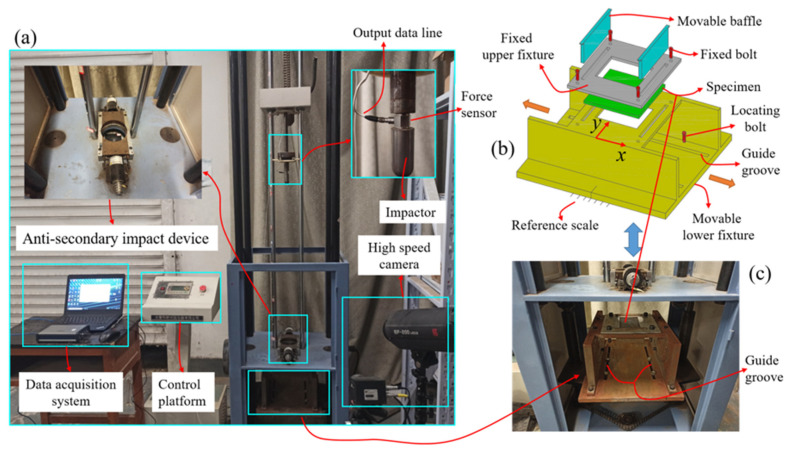
Dropweight impact test platform: (**a**) XBL-300 drop weight impact test machine, (**b**) improved specimen fixture and (**c**) photograph of improved specimen fixture.

**Figure 3 polymers-15-01403-f003:**
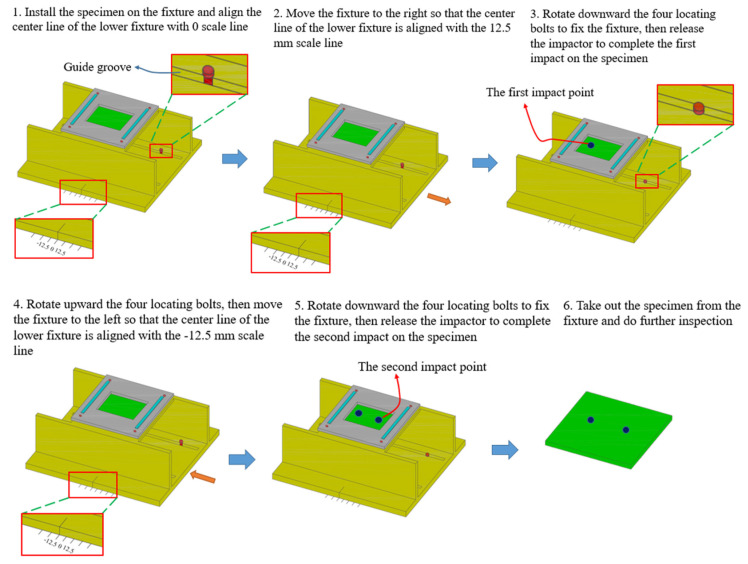
Low-velocity double-impacts test process.

**Figure 4 polymers-15-01403-f004:**
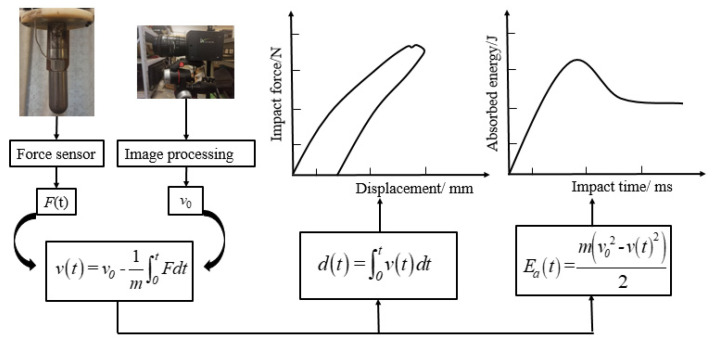
Flow chart of force–displacement curve and impact energy curve calculation.

**Figure 5 polymers-15-01403-f005:**
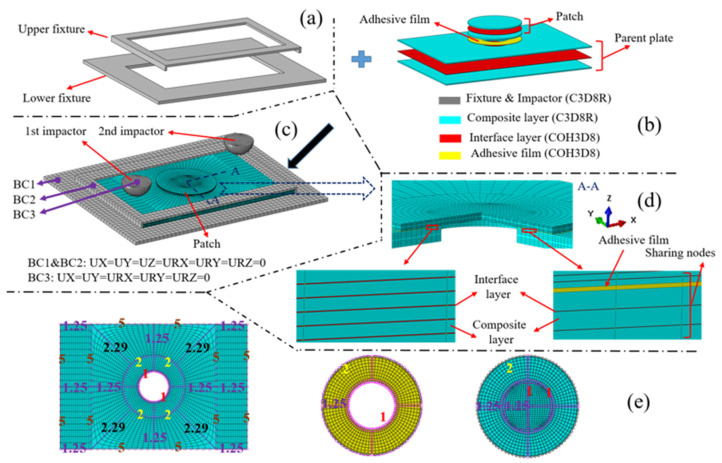
The FEM for double impacts: (**a**) fixtures, (**b**) components of the repaired specimen, (**c**) complete model and boundary conditions BC1-BC3, (**d**) section A-A with partial enlarged drawings in it and (**e**) seeds distribution for the edges of the parent plate, adhesive film and patch. 1-one seed per 1 mm, 2-one seed per 2 mm, 5-one seed per 5 mm.

**Figure 6 polymers-15-01403-f006:**
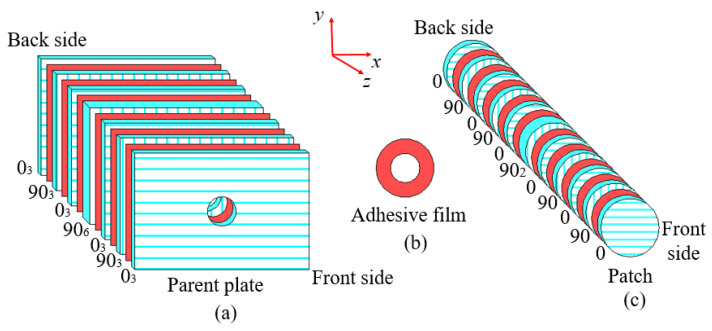
Composite layers and interface layers of (**a**) the parent plate, (**b**) patch and (**c**) adhesive film.

**Figure 7 polymers-15-01403-f007:**
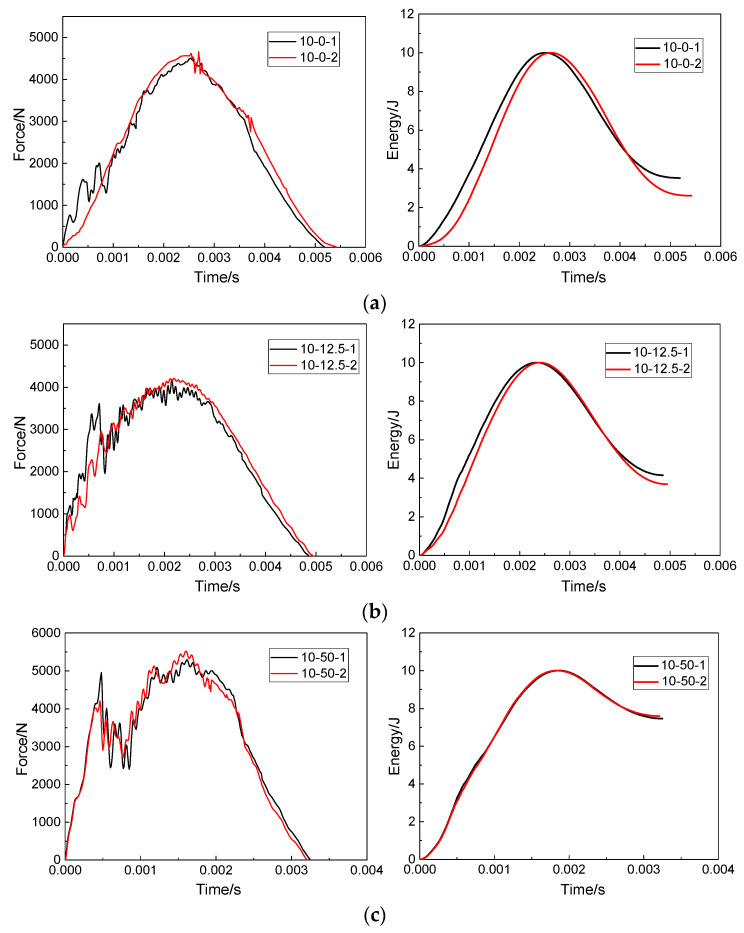
Force–time curves and impact energy curves of patch-repaired specimens under double impacts at impact distances of (**a**) 0 mm, (**b**) 12.5 mm, (**c**) 50 mm.

**Figure 8 polymers-15-01403-f008:**
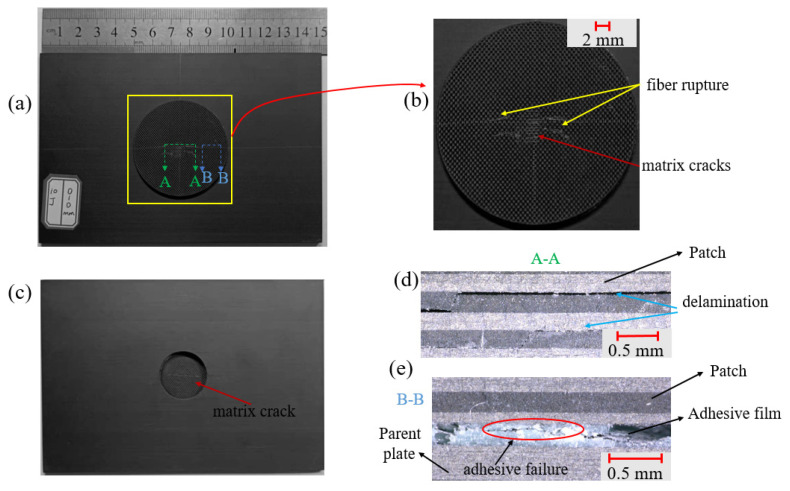
Micrographs of the patch-repaired specimen after the third impact at an impact distance of 0 mm: (**a**) the front part, (**b**) local figure of the patch, (**c**) the back part, (**d**) the side view part and (**e**) the A-A section.

**Figure 9 polymers-15-01403-f009:**
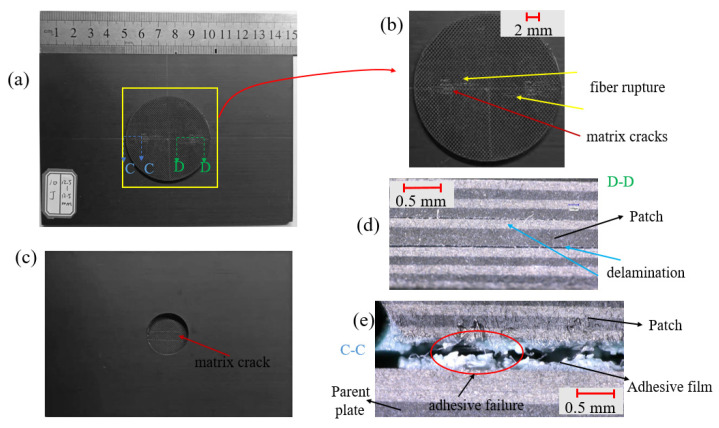
Micrographs of the patch-repaired specimen after the third impact at a distance of 12.5 mm: (**a**) the front part, (**b**) local figure of the patch, (**c**) the back part, (**d**) D-D section and (**e**) C-C section.

**Figure 10 polymers-15-01403-f010:**
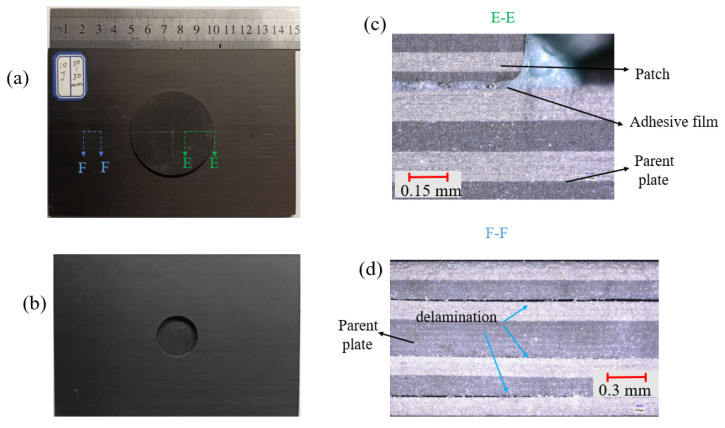
Micrographs of the patch-repaired specimen after the third impact at a distance of 50 mm: (**a**) the front part, (**b**) the back part, (**c**) E-E section and (**d**) F-F section.

**Figure 11 polymers-15-01403-f011:**
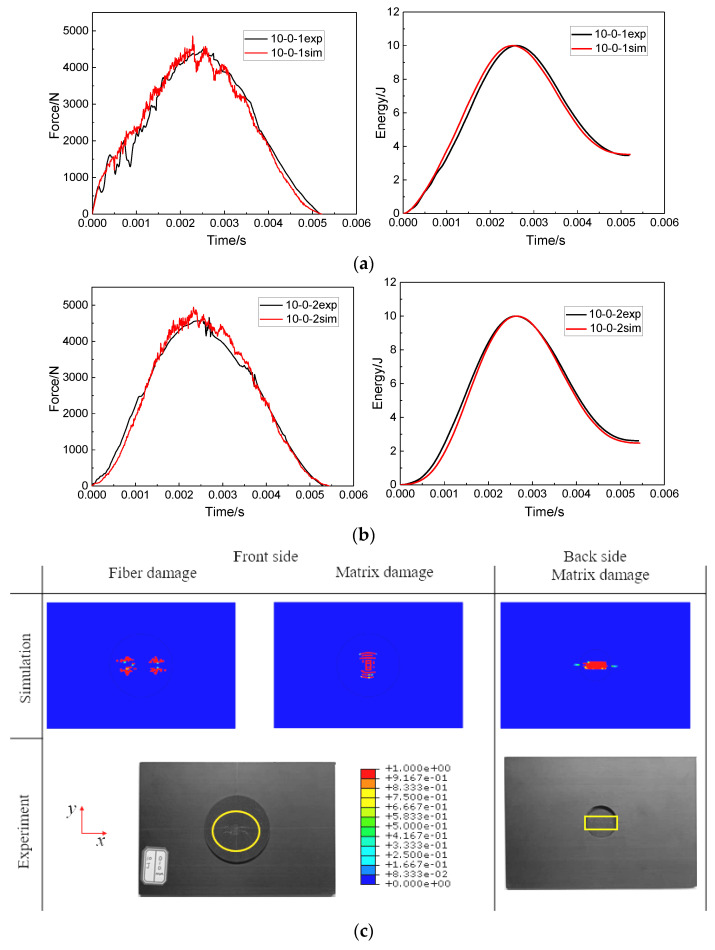
Comparisons of force–time curves and impact energy curves between the simulation and experiment results at an impact distance of 0 mm: (**a**) the first impact, (**b**) the second impact and (**c**) comparisons of the damage profiles after the second impact.

**Figure 12 polymers-15-01403-f012:**
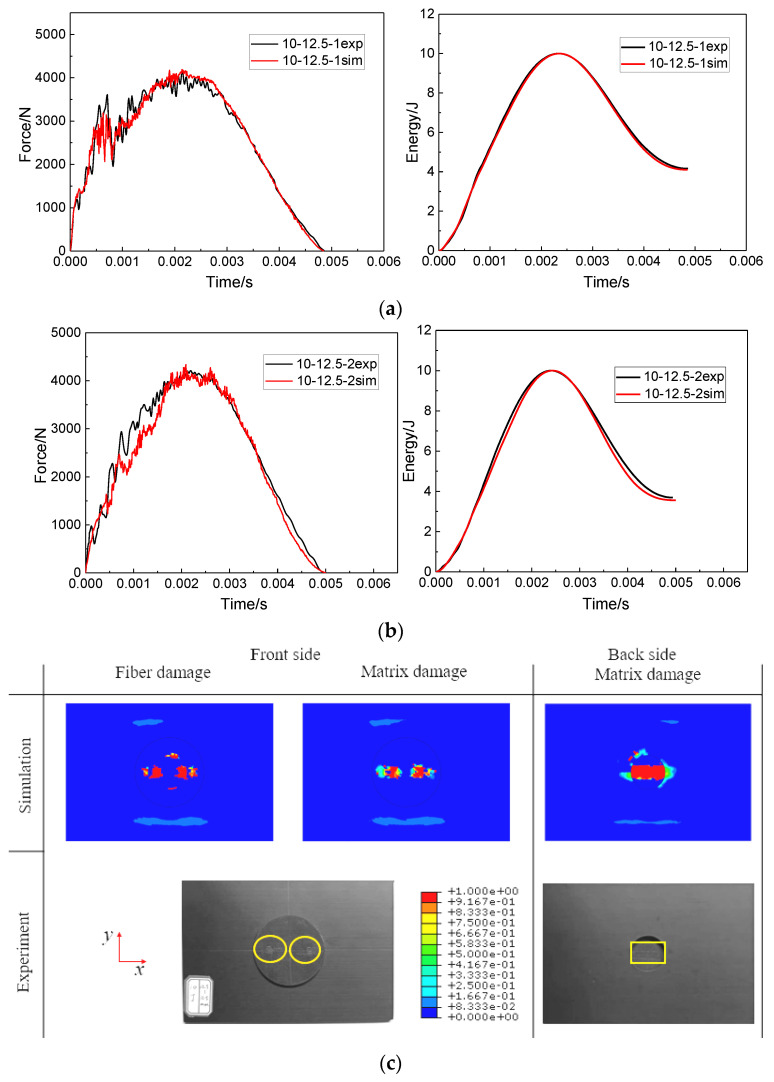
Comparisons of force–time curves and impact energy curves between the simulation results and the corresponding experiment results at an impact distance of 12.5 mm: (**a**) the first impact, (**b**) the second impact and (**c**) comparisons of the damage profiles after the second impact.

**Figure 13 polymers-15-01403-f013:**
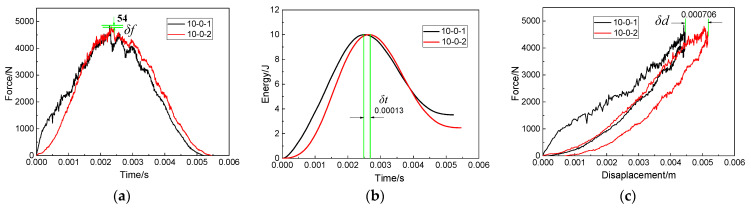
Force–time curves (**a,d,g,j,m**), absorbed energy curves (**b,e,h,k,n**) and force–displacement curves (**c,f,i,l,o**) under double impacts at the impact distances of 0 mm–50 mm, with the impact energy of 10 J.

**Figure 14 polymers-15-01403-f014:**
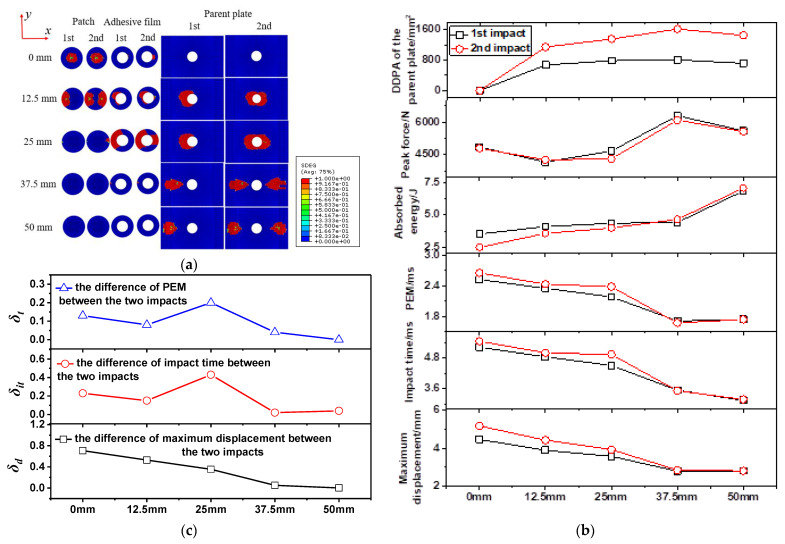
Delamination damage diagrams of the repaired specimen (**a**), impact parameter curves under double impacts at impact distances of 0 mm–50 mm, with the impact energy of 10 J, (**b**) and difference curves of impact parameters (**c**).

**Figure 15 polymers-15-01403-f015:**
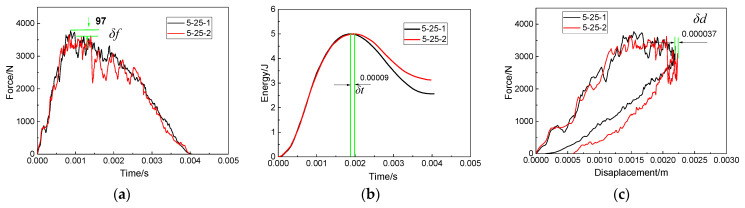
Force–time curves (**a,d,g,j,m**), absorbed energy curves (**b,e,h,k,n**) and force–displacement curves (**c,f,i,l,o**) under double impacts at the impact distance of 25 mm, with the impact energy of 5 J–15 J.

**Figure 16 polymers-15-01403-f016:**
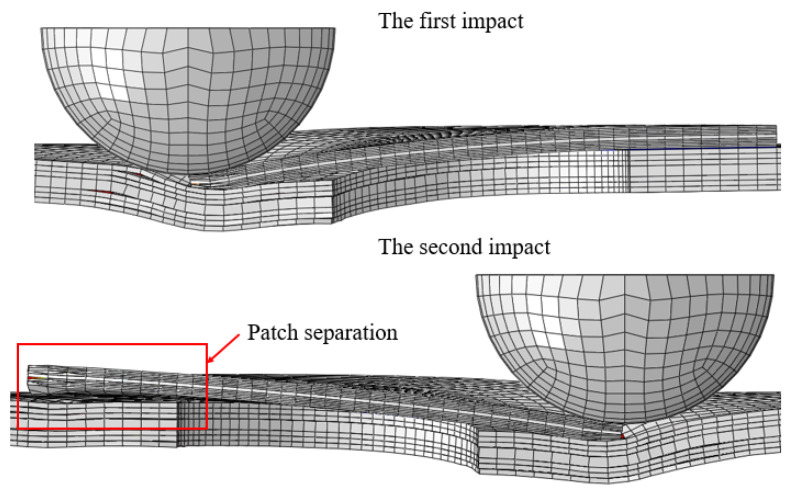
Contact between the impactor and the patch during the first impact and the second impact at the impact energy of 15 J.

**Figure 17 polymers-15-01403-f017:**
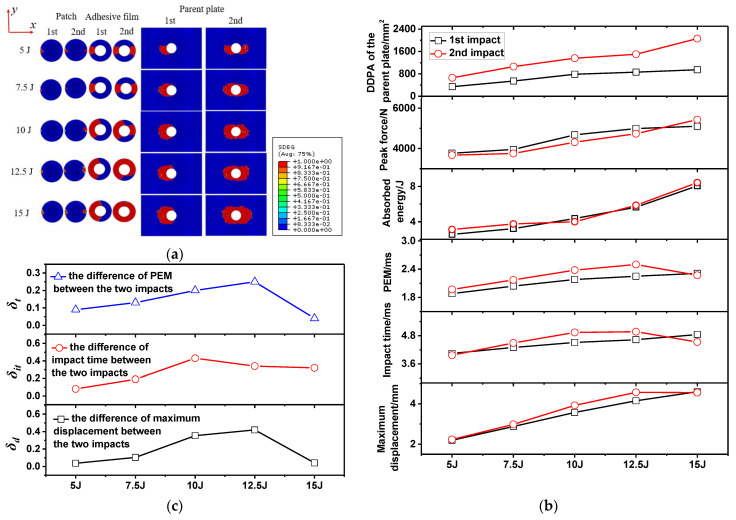
Delamination damage diagrams of the repaired specimen (**a**), impact parameter curves under double impacts at the impact distance of 25 mm, with the impact energy of 5 J–15 J, (**b**) and difference curves of the impact parameters (**c**).

**Table 1 polymers-15-01403-t001:** Impact test parameters of patch-repaired double-impacts specimens.

Item	Value
Weight of impactor/kg	2.5
Radius of impactor/mm	12.5
Impact energies/J	10
Initial impact velocities/(m/s)	2.828
Impact distance/mm	0, 12.5, 50

**Table 2 polymers-15-01403-t002:** Mechanical properties of the used T300/7901 patch-repaired CFRP laminates.

Item	T300/7901 Composites	Interface Layer of T300/7901	LJM-200 Adhesive Film
Modulus	E11 = 125 GPa, E22 = E33 = 11.3 GPa, G12 = G13 = 5.43 GPa, G23 = 3.979 GPa	Kn = Ks = Kt 100 GPa/mm	Kn = Ks = Kt 100 GPa/mm
Poisson’s ratio	ν12 = ν13 = 0.3, ν23 = 0.42	-	-
Strength	XT = 2000 MPa, XC = 1100 MPa, YT = 80 MPa, YC = 280 MPa, S12 = S13 = 120 MPa, S23 = 89.9 MPa	$\begin{array}{l} t_{n}^{0}=50\;\mathrm{Mpa}\\ t_{s}^{0}=t_{t}^{0}=90\;\mathrm{Mpa} \end{array}$	$\begin{array}{l} t_{n}^{0}=80\;\mathrm{Mpa}\\ t_{s}^{0}=t_{t}^{0}=146\;\mathrm{Mpa} \end{array}$
Fracture energy	$\begin{array}{l} G_{ft}^{C}=180\;N/\mathrm{mm},\;G_{fc}^{C}=100\;N/\mathrm{mm}\\ G_{mt}^{C}=4\;N/\mathrm{mm},\;G_{mc}^{C}=10\;N/\mathrm{mm} \end{array}$	$\begin{array}{l} G_{n}^{C}=0.52\;N/\mathrm{mm},\\ G_{s}^{C}=G_{t}^{C}=0.92\;N/\mathrm{mm} \end{array}$	$\begin{array}{l} G_{n}^{C}=0.52\;N/\mathrm{mm},\\ G_{s}^{C}=G_{t}^{C}=1.002\;N/\mathrm{mm} \end{array}$
Density	ρ = 1.478 g/cm^3^	ρ = 1.478 g/cm^3^	ρ = 1.5 g/cm^3^

**Table 3 polymers-15-01403-t003:** Comparisons of impact parameters at impact distances of 0 mm and 12.5 mm (“-” indicate minus).

Impact Distance		Peak Force/N		Impact Time/ms		Absorbed Energy/J		Maximum Displacement/mm	
		Exp.	Sim. Error/%	Exp.	Sim. Error/%	Exp.	Sim. Error/%	Exp.	Sim. Error/%
0 mm	1st impact	4500.4	4855.3 7.89	5.38	5.22 −2.97	3.46	3.52 1.73	4.523	4.456 −1.48
	2nd impact	4658.9	4801.3 3.06	5.41	5.45 0.74	2.61	2.47 −5.36	5.129	5.162 0.64
12.5 mm	1st impact	4077.8	4171.6 2.30	4.86	4.85 −0.21	4.16	4.10 −1.44	3.876	3.889 0.34
	2nd impact	4204.9	4271.7 1.59	4.94	5.00 1.21	3.69	3.56 −3.52	4.387	4.417 0.68

**Table 4 polymers-15-01403-t004:** Impact parameters under double impacts at impact distances of 0 mm–50 mm, with the impact energy of 10 J.

Simulations	*F*_P_/N	*d*_max_/mm	Impact Time/ms	PEM/ms	Absorbed Energy/J	DDPA/mm^2^		
						Patch	Adhesive Film	Parent Plate
10-0-1	4855.26	4.456	5.22	2.52	3.52	366	5	0
10-0-2	4801.28	5.162	5.45	2.65	2.47	529	61	0
10-12.5-1	4171.63	3.889	4.85	2.35	4.10	468	165	672
10-12.5-2	4271.72	4.417	5.00	2.43	3.56	997	188	1150
10-25-1	4678.39	3.572	4.51	2.18	4.36	6	589	788
10-25-2	4313.56	3.925	4.94	2.38	3.99	67	841	1360
10-37.5-1	6287.45	2.803	3.54	1.71	4.42	0	6	810
10-37.5-2	6090.67	2.854	3.52	1.67	4.66	0	7	1622
10-50-1	5603.09	2.813	3.14	1.74	6.89	0	0	720
10-50-2	5569.56	2.815	3.18	1.74	7.12	0	0	1454

**Table 5 polymers-15-01403-t005:** Impact parameters under double impacts at the impact distance of 25 mm, with the impact energy of 5 J–15 J.

Simulations	*F*_P_/N	*d*_max_/mm	Impact Time/ms	PEM/ms	Absorbed Energy/J	DDPA/mm^2^		
						Patch	Adhesive Film	Parent Plate
5-25-1	3770.30	2.191	4.05	1.88	2.56	49	168	343
5-25-2	3673.68	2.228	3.97	1.97	3.12	61	345	661
7.5-25-1	3955.16	2.876	4.30	2.04	3.21	13	289	543
7.5-25-2	3751.93	2.980	4.49	2.17	3.74	13	601	1060
10-25-1	4678.39	3.572	4.51	2.18	4.36	6	589	788
10-25-2	4313.56	3.925	4.94	2.38	3.99	67	841	1360
12.5-25-1	4991.37	4.149	4.63	2.25	5.64	47	681	862
12.5-25-2	4733.61	4.568	4.97	2.50	5.86	129	1083	1500
15-25-1	5107.50	4.602	4.85	2.31	8.06	76	710	947
15-25-2	5432.62	4.561	4.53	2.27	8.43	168	1470	2062

## Data Availability

Not applicable.

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
