# Peer review of "Experimental and Numerical Investigation on the Influence Factors of Damage Interference of Patch-Repaired CFRP Laminates under Double Impacts"

_polymers, 2023, doi:10.3390/polym15061403_

Round 1

Author Response

  1. The article lacks information on what method was used to make the CFRPs woven laminates that were pyrolyzed.

Response:

Thank you very much for your suggestion. The composite plate mentioned in the article is a common laminated composite plate, which is obtained from Shandong Weihai Composite Material Co., Ltd. The corresponding information about composite laminates is detailed in Section 2.1 of the original text. If you have any other questions about the materials used in the paper, please ask us in time.

  1. It is recommended that Figure 5 be modified and presented in 5 separate figures. This will be more readable.

Response:

Thank you very much for your suggestion. We have reclassified Figure 5 and displayed it as 5 separate figures in the revised manuscript.

  1. Has the effect of finite element mesh density on the accuracy of the results been investigated as part of the numerical simulation?

Response:

Thank you very much for your question. We built a finite element model in ABAQUS software to predict the low-speed impact performance of the repaired specimens. There is a close relationship between the density of finite element mesh and the accuracy of simulation results. The higher the density of finite element mesh, the closer the simulation results are to the test results, but the computational time spent in the computer also increases greatly. On the basis of comprehensive consideration of calculation time and simulation accuracy, after repeated adjustments, we finally determined the mesh density of each part of the finite element model. The detailed information can be obtained from Figure 5 (e) of the revised manuscript.

  1. It is advisable to explain in the paragraph above the figure what the 10-0/12.5/50-1 and 10-0/12.5/50-2 tests mean. This can be read from the text of the paper, but it is difficult?

Response:

Thank you very much for your suggestion.

In order to distinguish the experiments, a notation E-d-1/2 was used to represent a specific pair of experiment, where E represented the impact energy, d represents the distance from the impact point of the first impact (1) and the impact point of the second impact (2) to the center of the test specimen, respectively. It should be noted that two impact points were all at the midpoint of the specimen when d was taken as 0 mm.

We mentioned this naming method for the first time in Section 2.3 of the original manuscript. For the convenience of reading, we have repeated the naming method in Section 4.1 of the revised manuscript.

  1. It is proposed to modify figure 2 and to present the experimental position in several figures?

Response:

Thank you very much for your suggestion. We have modified Figure 2 to facilitate the reader to obtain the installation position of the test specimen in the experiment from Figure 2 (b) and Figure 2 (c).

  1. Regarding the object of research, it is worth mentioning in the introduction recent manuscripts presenting the advantages and limitations of modern structures made of reinforced polymer composites. I suggest reading the following papers presenting studies of plate elements made of CFRP.

10.1016/j.compstruct.2020.112502

10.3390/ma13132956

10.1016/j.compstruct.2021.114345.

Response:

Thank you for your suggestion. We have added relevant content to the Introduction.

Reviewer 2 Report

1. In introduction part add more current data in this section.

2. add some more data in conclusion section according to your results.

3. Refer recent journal papers.

Author Response

  1. In introduction part add more current data in this section.

Response:

Thank you for your suggestion. We have added relevant content to the Introduction.

  1. Add some more data in conclusion section according to your results.

Response:

Thank you for your suggestion. We have added relevant content to the Conclusion.

  1. Refer recent journal papers.

Response:

Thank you for your suggestion. We have added relevant content to the References.
